# Comparison of Four Systems for SARS-CoV-2 Antibody at Three Time Points after SARS-CoV-2 Vaccination

**DOI:** 10.3390/diagnostics12061349

**Published:** 2022-05-29

**Authors:** Jong Do Seo, Minjeong Nam, Tae Hwan Lee, Yeon-Sun Ahn, Seon-Hyeon Shin, Hye Young Han, Hee-Won Moon

**Affiliations:** 1Department of Laboratory Medicine, Konkuk University School of Medicine, Seoul 05030, Korea; akame84@hanmail.net (J.D.S.); davidtl@hanmail.net (T.H.L.); tmango@hanmail.net (Y.-S.A.); shyunie@kuh.ac.kr (S.-H.S.); calla@kuh.ac.kr (H.Y.H.); 2Department of Laboratory Medicine, Korea University Anam Hospital, Seoul 02841, Korea; blueccoma@gmail.com

**Keywords:** SARS-CoV-2, vaccination, antibody, assays

## Abstract

Background: Immunity against severe acute respiratory syndrome coronavirus 2 (SARS-CoV-2) wanes over time after vaccination. Methods: We compared SARS-CoV-2 antibody levels in serial samples from 350 vaccinated individuals at 3 time points (3 weeks after the first or second dose and before the third dose) with 4 assays: GenScript cPASS SARS-CoV-2 neutralization antibody detection kits (cPASS), Siemens SARS-CoV-2 IgG (sCOVG), Abbott SARS-CoV-2 IgG II Quant (CoV-2 IgG II), and an Immuno-On™ COVID-19 IgG test (Immuno-On IgG). Antibody levels by time, concordance between assays, and values from other tests corresponding to the percent inhibition results in cPASS were assessed. Results: The median values at three time points were 49.31%, 90.87%, and 53.38% inhibition for cPASS, 5.39, 13.65, and 2.24 U/mL for sCOVG, 570.25, 1279.65, and 315.80 AU/mL for CoV-2 IgG II, and 223.22, 362.20, and 62.20 relative units (RU) for Immuno-On IgG. The concordance with cPASS at each time point ranged from 0.735 to 0.984, showing the highest concordance in the second sample and lowest concordance in the third in all comparative tests. The values corresponded to 30% inhibition, and the cutoffs of cPASS, were 2.02 U/mL, 258.6 AU/mL, and 74.2 RU for each test. Those for 50%, 70%, and 90% inhibition were 3.16, 5.66, and 8.26 U/mL for sCOVG, while they were 412.5, 596.9, and 1121.6 AU/mL for CoV-2 IgG II and 141.8, 248.92, and 327.14 RU for Immuno-On IgG. Conclusions: This study demonstrated the dynamic changes in antibody values at different time points using four test systems and is expected to provide useful baseline data for comparative studies and standardization efforts in the future.

## 1. Introduction

Coronavirus disease 2019 (COVID-19), caused by severe acute respiratory syndrome coronavirus 2 (SARS-CoV-2), was first reported in 2019 and has spread worldwide [1]. Despite the development and introduction of diverse vaccines such as BNT162b2 developed by BioNTech/Pfizer, mRNA-1273 developed by Moderna (Cambridge, MA, USA), and ChAdOx1 nCoV-19 developed by Oxford/AstraZeneca [2], as well as their confirmed efficacy for protection from the disease [3,4,5,6,7,8], the COVID-19 pandemic is still ongoing [1]. The waning of humoral immunity and the decrease in antibodies against SARS-CoV-2 induced by vaccination over time have been revealed in previous studies [9,10,11]. As defense against infection is correlated with the antibody titer [12,13], assays for anti-SARS-CoV-2 antibody detection and quantification can be helpful tools for the evaluation of disease protection. Several laboratory assays using diverse test principles and targeting various antibodies such as RBD-binding or neutralizing antibodies against SARS-CoV-2 have been developed and launched [14]. Tests evaluating virus neutralization using a pseudovirus-based virus neutralization test [15,16] or surrogate virus neutralization test (sVNT) [17,18] directly assess humoral immunity. However, there are restrictions to automation and increasing test throughput due to manual processing and incubation time. Automated immunoassays targeting RBD-binding antibodies have also been developed as alternative indicators of humoral immunity [14,19,20,21,22]. The titer of the binding antibody correlates with the neutralization of the neutralizing antibody [23,24]. Therefore, it is expected to be an alternative indicator for semi-automated and semiquantitative neutralization tests.

In this study, antibody tests were performed in four test systems for serum samples obtained from vaccinated individuals at three time points over a period of 6 months after completion of the second dose. The change in antibody levels over time, the qualitative decision, and the measured values were compared between samples from different time points using each test system. In addition, the quantitative or semiquantitative values from the binding antibody assays that correspond to each point of percent signal inhibition on neutralization test, including 30% signal inhibition of the test cutoff, are estimated in this study.

## 2. Materials and Method

### 2.1. Samples

This study recruited 380 subjects who were vaccinated with BNT162b2 or ChAdOx1 nCoV-19 at our institution. All subjects were ≥18 years of age at enrollment. Vaccination was performed according to the manufacturer’s instructions, and blood samples were drawn from each subject at three time points: 3 weeks after the first dose, 3 weeks after the second dose, and before the third dose from March to November 2021. In accordance with the government’s guidelines for vaccination, the dosing interval between the first and second booster doses of BNT162b2 was 3 weeks, and that of ChAdOx1 nCoV-19 ranged from 8 to 12 weeks. The interval between the second and third booster doses of BNT162b2 was 6 months, and that of ChAdOx1 nCoV-19 was 5 months. Among the 380 participants initially included in the study, those whose blood samples were not taken, patients who were diagnosed with COVID-19, and individuals who had received booster vaccination before the third blood collection were excluded from the analysis. A total of 350 subjects remained, and 348, 318, and 264 samples were collected for each respective time point. For cPASS, the indicators of neutralization were confirmed for 183, 314, and 253 samples from each point, which were included for comparison analysis, whereas some samples with sCOVG and CoV-2 IgG II results not paired with cPASS were included in the result distribution analysis only. As described in a previous study [24], the blood samples were collected into Vacuette CAT serum clot activator (Greiner Bio-One, Kremsmunster, Austria) and centrifuged at 1977× *g* for 10 min, and the serum was aliquoted into two microcentrifuge tubes and stored at −80 °C until measured in four test systems as described below.

### 2.2. Assays

The samples were evaluated using four test systems. The results of the sVNT—a GenScript cPASS SARS-CoV-2 neutralization antibody detection kit (cPASS; GenScript, USA Inc., Piscataway, NJ, USA) adopting an enzyme-linked immunosorbent assay (ELISA) as a test principle—were used as the standard for virus neutralization in the present study. Siemens SARS-CoV-2 IgG (sCOVG; Siemens Healthcare Diagnostics Inc., Tarrytown, NY, USA) and Abbott SARS-CoV-2 IgG II Quant (CoV-2 IgG II; Abbott Laboratories, Sligo, Ireland) were selected as representative automated quantitative binding antibody immunoassays, with an Immuno-On™ COVID-19 IgG test (Immuno-On IgG; Osang Healthcare Inc., Anyang, Korea) for semiquantitative lateral flow immunoassays. All the tests were performed in accordance with the manufacturer’s instructions.

### 2.3. Data Analysis

The distribution and change in test results for each system over time were analyzed, and the decisions and values of each system were compared with those of the neutralization test. The concordance of decision with the neutralization test and its 95% confidence interval (CI) for each time point and the entire period was analyzed by diagnostic test evaluation, using the neutralization test as the gold standard. The optimal cutoff to obtain the maximum diagnostic accuracy corresponding to 30%, 50%, 70%, and 90% signal inhibition in cPASS was determined using receiver operating characteristic (ROC) curve analysis by calculating the area under the ROC curve (AUC) and a 95% CI. All statistical analyses were performed using IBM SPSS Statistics for Windows (version 26.0; IBM Corp., Armonk, NY, USA) and MedCalc version 14.8.1 (MedCalc Software, Ostend, Belgium).

## 3. Results

The seropositive or negative test rates were determined by the manufacturer-claimed cutoff of each test system: ≥30% signal inhibition for cPASS, ≥1.0 U/mL for sCOVG, ≥50.0 AU/mL for CoV-2 IgG II, and ≥14.9 RU for Immuno-On IgG. As shown in Table 1 and Figure 1, the results from the four test systems showed a maximal increase in antibody levels after the second dose of vaccination, which decreased over time. For the cPASS test, the median and interquartile range (IQR) of the percent inhibition was 49.31% (31.79–65.99%) at 3 weeks after the first dose, which then increased to 90.87% (75.66–96.53%) at 3 weeks after the second dose and decreased to 53.38% (31.82–75.18%) before the third dose. The seropositivity rates were 79.8%, 97.8%, and 77.9% at each timepoint. For sCOVG, the median and IQR were 5.39 U/mL (2.28–12.90 U/mL), 13.65 U/mL (6.97–53.17 U/mL), and 2.24 U/mL (1.25–4.94 U/mL), respectively, showing a similar change to the neutralization test, an increase after the second dose, and a decrease over time. The seropositivity rates according to the manufacturer-claimed cutoffs were 90.4%, 99.7%, and 81.3%, respectively. For the CoV-2 IgG II test, the median and IQR were 570.25 AU/mL (252.45–1308.20 AU/mL), 1279.65 AU/mL (714.00–3764.60 AU/mL), and 315.80 AU/mL (181.40–682.50 AU/mL), and the seropositive rates were 96.2%, 99.7%, and 96.6%, respectively. For the Immuno-On IgG test, the median and IQR were 223.22 relative units (RU) (56.71–347.30 RU), 362.20 RU (232.86–456.80 RU), and 62.20 RU (17.39–213.40 RU), and the seropositive rates were 88.0%, 99.4%, and 79.1% at each point, respectively.

The concordance of qualitative decisions between cPASS and other assays at three time points ranged from 0.897 to 0.981 for sCOVG, from 0.814 to 0.981 for CoV-2 IgG II, and from 0.735 to 0.984 for Immuno-On IgG, as shown in Figure 2. The trend in the highest concordance in the second sample and lowest concordance in the third sample was observed in all comparative tests. The concordance with cPASS for all samples from the entire study period was 0.935 for sCOVG, 0.891 for CoV-2 IgG II, and 0.879 for Immuno-On IgG.

In the ROC curve analysis based on the decision of cPASS with a claimed cutoff of 30% inhibition for all samples, the area under the curve (AUC) and its 95% CI were 0.962 (0.945–0.974) for sCOVG, 0.963 (0.947–0.976) for CoV-2 IgG II, and 0.845 (0.817–0.870) for Immuno-On IgG. In contrast to the high sensitivity of 94.8–100% at the claimed cutoff, the specificity showed relatively low values of 19.0–59.0%, as shown in Table 2. The optimal cutoff to obtain the highest diagnostic accuracy was ≥2.02 U/mL for sCOVG, ≥258.6 AU/mL for CoV-2 IgG II, and ≥74.2 RU for Immuno-On IgG, which were much higher than the manufacturer’s suggestion. When the optimal cutoff was applied to the analysis, the sensitivity of the three systems ranged from 80.0% to 88.2%, and the specificity ranged from 80.0% to 95.0%. The increase in the specificity surpassed the decrease in sensitivity, and thus the total diagnostic accuracy improved.

Similar to the values from the binding antibody assays that corresponded to the estimated 30% inhibition of cPASS, the values that corresponded to 50%, 70%, and 90% inhibition were calculated, and the qualitative decision concordance with the AUC in the ROC curve analysis for 30% inhibition and other alternate cutoff values were estimated. The values that corresponded to 50%, 70%, and 90% inhibition were 3.16, 5.66, and 8.26 U/mL for sCOVG, 412.5, 596.9, and 1121.6 AU/mL for CoV-2 IgG II, and 141.8, 248.92, and 327.14 RU for Immuno-On IgG, respectively. The decision concordance following the change in the percent inhibition cutoff is shown in Appendix A, and the AUC and 95% CI of each system, which corresponded to 50%, 70%, and 90% inhibition, were 0.941 (0.922–0.957), 0.911 (0.888–0.930), and 0.936 (0.916–0.953) for sCOVG, 0.944 (0.925–0.959), 0.917 (0.895–0.936), and 0.940 (0.920–0.956) for CoV-2 IgG II, and 0.867 (0.840–0.890), 0.863 (0.836–0.887), and 0.899 (0.875–0.920) for Immuno-On, respectively. The sensitivity and specificity of the binding antibody assays at the claimed or newly estimated cutoff values compared with the diverse cPASS cutoffs are shown in Appendix A.

## 4. Discussion

sVNT is an available option for evaluation of the humoral immune response against SARS-CoV-2 induced by COVID-19 infection or vaccination [17,18]. This test, using ELISA as the test principle, is useful to evaluate virus neutralization and infection defense. However, there are obstacles to automated, massive, and rapid testing due to a long turn-around time and the manual process involved in the test. Therefore, if automated antibody tests that are appropriate for massive and rapid tests can be an alternative indicator of virus neutralization, it will become easier to meet the increasing demand for immunity evaluation. According to previous studies, protection against infection is known to be related to the titer of neutralizing antibodies [9], and the humoral immune response acquired by a vaccine is enhanced by the second dose [10,25]. Then, the titers of the neutralizing and binding antibodies wane over time [9,10,11], just as with immunity acquired through COVID-19 infection or immunity by vaccine for other infectious diseases [26,27]. In the present study, the effect of vaccination on humoral immunity and the change in immune response over time, which were identified by previous studies, were reconfirmed by four test systems adopting different test principles and detecting different targets. The distribution of the measured values showed statistically significant differences between the first, second, and third blood samples in the four test systems. The concordance of decision between the test systems showed the highest concordance in the decision for the second blood samples, which were expected to have the highest antibody titer, and showed the lowest concordance for the third blood samples, which were expected to have the lowest titer. The difference in concordance by the time of sample collection was considered to be due to the difference in antibody titer, and it is believed that the discrepancy largely originated from the difference between the performance of the systems for the decision of the low-concentration samples. The value of this study is that there were few studies showing the assessment of concordance by the time point of sample collection taken from the same subjects. From the early period of test development, there were attempts to evaluate the humoral immunity at a specific time point [28] or serial change over time by comparing multiple tests including our previous study [24]. However, few studies had evaluated the change up to just before the third dose of a vaccine. According to the results of this study, the timing of blood sampling after infection or vaccination and the expected antibody titer should be considered important factors in future studies to evaluate the concordance between antibody assays.

In the ROC curve analysis, each test showed high sensitivity and relatively low specificity when the claimed cutoff was applied, and the claimed cutoff was lower than the optimal cutoff to obtain maximum diagnostic accuracy. Therefore, the claimed cutoff is considered strict to secure high sensitivity and is appropriate for detecting seropositive subjects in a population with a high seropositive rate. Meanwhile, because of the high false-positive rate predicted in populations with low seropositive rates due to relatively low specificity, there are limitations on the use of seronegative subjects [14,22,29]. Since there is controversy about the standard for evaluating virus neutralization [13,30], the cutoff is not an absolute value but is changeable, similar to the change in the claimed cutoff for cPASS from 20–30% [24]. The values from other tests have been estimated in this study, corresponding to each point of signal inhibition in sVNT. The predicted values corresponding to 30% inhibition—2.02 U/mL for sCOVG and 258.6 AU/mL for CoV-2 IgG II—were slightly lower than those of our previous study—2.42 U/mL and 284.0 AU/mL [24]—estimated by the samples from only the first and second time points. The inclusion of samples from the third time point with low antibody titer and a relatively large decrease in binding antibodies compared with that of neutralizing antibodies are thought to be the cause of these results. As the cutoff of sVNT increased from 30% to 90% inhibition, more “positive” results have been reclassified as “negative,” and the concordance of decision with other tests with fixed cutoffs has decreased.

Through this study, high qualitative concordance with the neutralization test was confirmed in all the test systems included in present study, and the availability of an anti-SARS-CoV-2 antibody automation test and lateral flow immunochromatography as alternative indicators of the neutralization test has been evaluated by calculating the measured values in these platforms that correspond to 30%, 50%, 70%, and 90% signal inhibition in sVNT. It is considered that the relatively low concordance and AUC value in flow immunochromatography, compared with that of the quantitative binding antibody assays, are due to the characteristics of the test design, which was developed as a semiquantitative assay.

The present study has some limitations. First, it was impossible to include seronegative subjects as counterparts to the vaccinated subjects in the study design because of the continuous increase in the vaccination rate. As a result, a biased distribution of the decision occurred, and the Cohen–Kappa coefficient, assuming an even distribution between decisions, was affected [31]. Therefore, the coefficient for the comparison between test methods became meaningless and unavailable, and it was then excluded from the analysis. Second, the medical information of individuals could not be reviewed, because informed consent does not include the right to access the medical records of the subjects. Thus, data that were inappropriate for analysis due to COVID-19 infection after vaccination or an inaccurate booster vaccination schedule could not be completely excluded. Lastly, since this study was designed to include only vaccinated individuals, the possibility of limited distribution in humoral immunity or antibody titer cannot be excluded, despite the fact that individuals who received different vaccines such as BNT162b2 or ChAdOx1 nCoV-19, which are known for different antibody titer distributions, were enrolled for analysis. Therefore, it is necessary to note that there may be limitations in applying the results of the present study to the interpretation of test results in the clinical setting.

Despite these limitations, to the best of our knowledge, this is the first study to evaluate the dynamic changes in the humoral immune response against COVID-19 at three time points using four test systems. Therefore, this study is expected to provide useful baseline data for comparative study in the future. Moreover, these data could be helpful for the interpretation of humoral immunity using each binding antibody assay by showing the values of each test corresponding to various levels of percent inhibition in the neutralization test. Efforts to standardize the values of the diverse antibody tests are required in future studies.

## Figures and Tables

**Figure 1 diagnostics-12-01349-f001:**
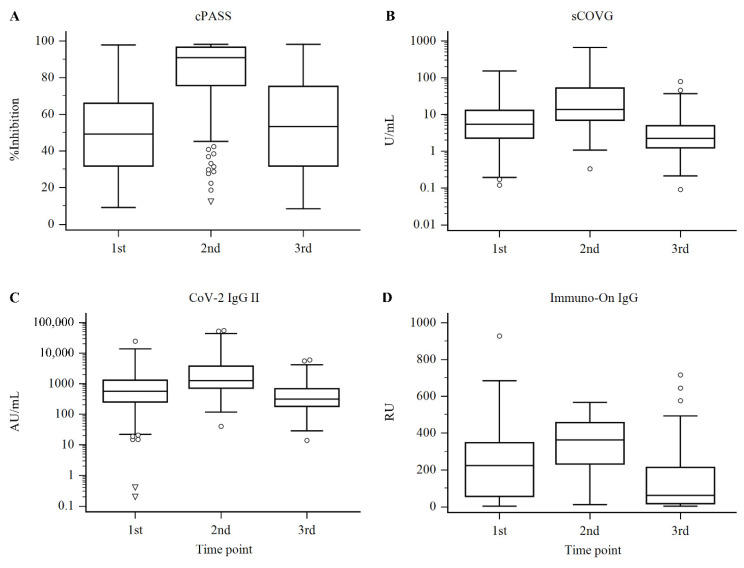
The measured results of the samples for each test system: (**A**) percent inhibition for cPASS, (**B**) U/mL for sCOVG, (**C**) AU/mL for CoV-2 IgG II, and (**D**) RU for Immuno-On IgG, presented as box and whisker plots. The horizontal line and the box represent the median and interquartile range, respectively, whereas the circle and inverted triangles represent near and far outliers confirmed by Tukey’s method. The results for sCOVG and CoV-2 IgG II are presented as log scales.

**Figure 2 diagnostics-12-01349-f002:**
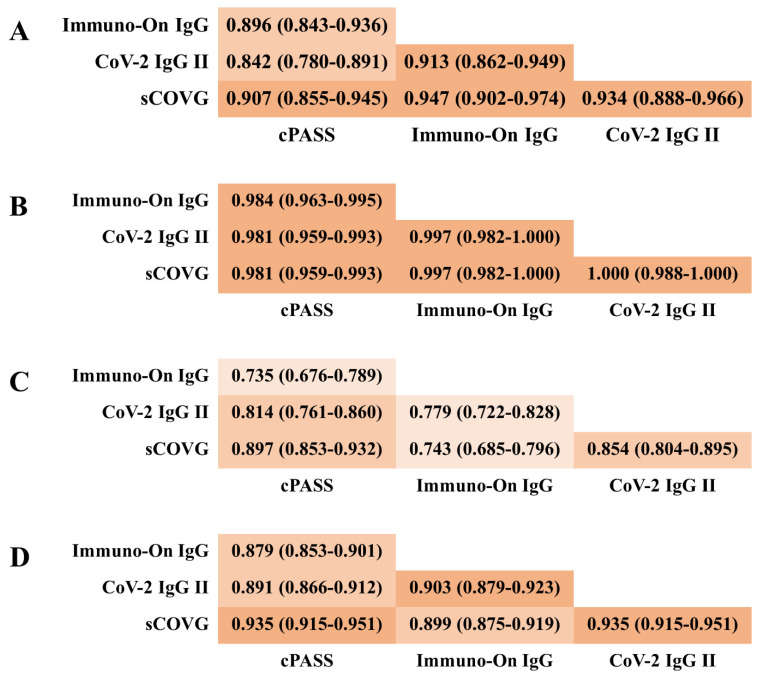
Concordance and 95% CI of qualitative decision between test systems for serum samples collected at (**A**) 3 weeks after the first dose, (**B**) 3 weeks after the second dose, (**C**) 6 months after the second dose, and (**D**) the whole samples. If there is no overlap of 95% CIs, it can be interpreted as having a statistically significant difference at a significant level of 0.05 (5%).

**Table 1 diagnostics-12-01349-t001:** The median and interquartile range of measured value and decision on humoral immunity over time measured by multiple test systems.

System	Sample	n	Median and IQR	Decision *	
				Positive	Negative
cPASS	First	183	49.31 (31.79–65.99)	79.8% (146/183)	20.2% (37/183)
(% inhibition)	Second	315	90.87 (75.66–96.53)	97.8% (308/315)	2.2% (7/315)
	Third	253	53.38 (31.82–75.18)	77.9% (197/253)	22.1% (56/253)
sCOVG	First	344	5.39 (2.28–12.90)	90.4% (311/344)	9.6% (33/344)
(U/mL)	Second	315	13.65 (6.97–53.17)	99.7% (314/315)	0.3% (1/315)
	Third	262	2.24 (1.25–4.94)	81.3% (213/262)	18.7% (49/262)
CoV-2 IgG II	First	344	570.25 (252.45–1308.20)	96.2% (331/344)	3.8% (13/344)
(AU/mL)	Second	316	1279.65 (714.00–3764.60)	99.7% (315/316)	0.3% (1/316)
	Third	262	315.80 (181.40–682.50)	96.6% (253/262)	3.4% (9/262)
Immuno-On IgG	First	183	223.22 (56.71–347.30)	88.0% (161/183)	12.0% (22/183)
(RU)	Second	315	362.20 (232.86–456.80)	99.4% (313/315)	0.6% (2/315)
	Third	253	62.20 (17.38–213.40)	79.1% (200/253)	20.9% (53/253)

The decisions were determined by the manufacturer-claimed cutoff of each test system: ≥30% signal inhibition for cPASS, ≥1.0 U/mL for sCOVG, ≥50.0 AU/mL for CoV-2 IgG II, and ≥14.9 RU for Immuno-On IgG. * The seropositive or negative rates are presented as percentages with the number of subjects in parentheses. Abbreviations: IQR = interquartile range; cPASS = GenScript cPASS SARS-CoV-2 neutralization antibody detection kit; sCOVG = Siemens SARS-CoV-2 IgG; CoV-2 IgG II = Abbott SARS-CoV-2 IgG II Quant; Immuno-On IgG = Osang Immuno-On™ COVID-19 IgG test; RU = relative units.

**Table 2 diagnostics-12-01349-t002:** The sensitivity and specificity of each test system with trade-offs in the cutoff point, compared with the cPASS decision with a cutoff of 30% inhibition.

	AUC (95% CI *)	Cutoff	Trade-off	Sensitivity (95% CI)	Specificity (95% CI)
sCOVG	0.962 (0.945–0.974)	Claimed	≥1.0 U/mL	98.6 (97.4–99.4)	59.0 (48.7–68.7)
		Optimal	≥2.02 U/mL	87.2 (84.4–89.7)	94.0 (87.4–97.8)
CoV-2 IgG II	0.963 (0.947–0.976)	Claimed	≥50.0 AU/mL	100.0 (99.4–100.0)	19.0 (11.8–28.1)
		Optimal	≥258.6 AU/mL	88.2 (85.4–90.6)	95.0 (88.7–98.4)
Immuno-On IgG	0.845 (0.817–0.870)	Claimed	≥14.9 RU	94.8 (92.8–96.4)	44.0 (34.1–54.3)
		Optimal	≥74.2 RU	80.0 (76.7–83.0)	80.0 (70.8–87.3)

The sensitivity and specificity of the test systems at each cutoff point were determined based on the decision of cPASS with a cutoff of 30% inhibition. * If there is no overlap in the 95% CIs between parameters, it can be interpreted as having a statistically significant difference at the significance level of 0.05 (5%). Abbreviations: AUC = area under the curve; CI = confidence interval; cPASS = GenScript cPASS SARS-CoV-2 neutralization antibody detection kit; sCOVG = Siemens SARS-CoV-2 IgG; CoV-2 IgG II = Abbott SARS-CoV-2 IgG II Quant; Immuno-On IgG = Osang Immuno-On™ COVID-19 IgG test.

## Data Availability

A data set of serological responses of 930 samples was deposited at https://dataverse.harvard.edu/ (accessed on 25 February 2022) (https://doi.org/10.7910/DVN/CMRD77 accessed on 25 February 2022).

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
