# Peer review of "Comparison of Four Systems for SARS-CoV-2 Antibody at Three Time Points after SARS-CoV-2 Vaccination"

_diagnostics, 2022, doi:10.3390/diagnostics12061349_

Round 1

Reviewer 1 Report

The authors present an interesting dataset about Sars-CoV-2 antibody levels at 3 different time points after vaccination. The study compares 4 different methods to determine the IgG levels. Moreover, this study showed how such IgG levels could be used to predict the level of inhibition of the infection. The neutralization antibody detection tests are expensive and not compatible with a large-scale input.  The manuscript overall presents a lot of useful data than can add value COVID-19 diagnostic and prognostic field after some minor revisions that you can find in the attached file.

Author Response

Point 1: Need formatting

Response 1: The detailed formatting like font size, would be the job of editorial department.

Point 2: It would be usefull to specifiy the IgG level cut-off for each method beofre describe the percentage.

Response 2: The cutoff values of each system have been added to main text of revised manuscript.

Point 3: I think this paragraph should go before you describe the different percentages. My attention for example goes to the seropositivity rates after the 3rd doses. sCOVG and Immuno-On IgG have surprising lower %.

Response 3: It has been added to the main text in revised version, by accepting the reviewer's advice.

The third sample has been drawn just before the third vaccine dose, about 5-6 months later after the second dose.

Point 4: 30% inhibition cut-off has ti be specifiy also here, not only in the table legend.

Response 4: The reviewer's suggestion has been reflected in revised manuscript.

Point 5: More references on immune response fading over time also for infected and orvaccinated. De Carlo, A.; Lo Caputo, S.; Paolillo, C.; Rosa, A.M.; D’Orsi, U.; De Palma, M.; Reveglia, P.; Lacedonia, D.; Cinnella, G.; Foschino, M.P.; Margaglione, M.; Mirabella, L.; Santantonio, T.A.; Corso, G.; Dattoli, V. SARS-COV-2 Serological Profile in Healthcare Professionals of a Southern Italy Hospital. Int. J. Environ. Res. Public Health 2020, 17, 9324

Saffar, H., Mousavi, S., Saffar, H. et al. Seroconversion rates following 2 doses of measles- mumps- rubella vaccination given at the ages 12 and 18 months: data for possible additional dose at older age. BMC Immunol 23, 2 (2022).

Response 5: Those studies have been added to reference in revised manuscript.

Point 6: tables s1 doesn't fit the page. Cna't see the 90% values

Response 6: The landscape style in the Word file had not been applied to the PDF file for the reviewer. Please find the revised Word file we submit here together.

Reviewer 2 Report

The work presented for review concerns the comparison of serological tests detecting antibodies against SARS-CoV-2. Despite the fact that the topic of the work is not innovative, the authors analyzed the obtained results in an original way. The Introduction and the Materials and Methods are short, but in my opinion sufficient. Results well presented. After taking into account minor revision, the work should be published in Diagnostics.

Minor revision:

1)    Discussion - There are several papers by other authors that compare different serological tests in the vaccine response against SARS-CoV-2. These results should be analyzed in the discussion.

2)    The authors write about the statistical significance of the results. They also list the programs used for the statistical analysis. There is no clear information on what specific statistical tests were used.

3)    Materials and Methods - please write whether the given serological tests were carried out in accordance with the manufacturer's instructions or not. If not, please describe the modifications.

4)    Figure 1 - For the graph panel, the "Time point" captions at points A and B can be removed. The panel will be easier to read.

Author Response

Point 1: Discussion - There are several papers by other authors that compare different serological tests in the vaccine response against SARS-CoV-2. These results should be analyzed in the discussion.

Response 1: The description about the findings from other previous studies, change of immunity over time, comparison between assays, have been added to discussion in revised manuscript.

Point 2: The authors write about the statistical significance of the results. They also list the programs used for the statistical analysis. There is no clear information on what specific statistical tests were used.

Response 2: The description of ‘2.3. data analysis’ section has been reinforced in reviesd version.

Point 3: Materials and Methods - please write whether the given serological tests were carried out in accordance with the manufacturer's instructions or not. If not, please describe the modifications.

Response 3: The tests have been performed according to manufacturer’s instruction, and it stated in revised manuscript, ‘2.2 assyas’ section.

Point 4: Figure 1 - For the graph panel, the "Time point" captions at points A and B can be removed. The panel will be easier to read.

Response 4: The captions were removed for readability, by accepting the reviewer's advice.
